

# Realistic nitrate concentrations diminish reproductive indicators in *Skiffia lermae*, an endemic species in critical endangered status

Ivette Marai Villa-Villaseñor[1], Ma. Antonia Herrera-Vargas[2], Beatriz Yáñez-Rivera[3], Mari Carmen Uribe[4], Rebeca Aneli Rueda-Jasso[5], Bryan V. Phillips-Farfán[6], Valentin Mar-Silva[7], Esperanza Meléndez-Herrera[2] and Omar Domínguez-Domínguez[5]

[1] Programa Institucional de Doctorado en Ciencias Biológicas, Universidad Michoacana de San Nicolás de Hidalgo, Morelia, Michoacán, Mexico
[2] Laboratorio de Ecofisiología Animal, Instituto de Investigaciones sobre Recursos Naturales, Universidad Michoacana de San Nicolás de Hidalgo, Morelia, Michoacán, Mexico
[3] Unidad Académica Mazatlán, Instituto de Ciencias del Mar y Limnología, Universidad Nacional Autónoma de México, Mazatlán, Sinaloa, Mexico
[4] Laboratorio de Biología de la Reproducción Animal, Departamento de Biología Comparada, Facultad de Ciencias, Universidad Nacional Autónoma de México, Ciudad de México, Ciudad de México, Mexico
[5] Laboratorio de Biología Acuática, Facultad de Biología, Universidad Michoacana de San Nicolás de Hidalgo, Morelia, Michoacán, Mexico
[6] Laboratorio de Nutrición Experimental, Instituto Nacional de Pediatría, Ciudad de México, Mexico
[7] Estancia Posdoctoral por México-CONACyT, Escuela Nacional de Estudios Superiores Unidad Morelia, Universidad Nacional Autónoma de México, Morelia, Michoacán, Mexico

Corresponding author
Esperanza Meléndez-Herrera, emelendez@umich.mx

## ABSTRACT

Goodeinae is a subfamily of critically endangered fish native to central Mexico. Populations of *Skiffia lermae*, a species belonging to this subfamily, have significantly decreased in the past two decades. A previous study showed that *S. lermae* is sensitive to acute nitrate-nitrogen ($NO_3$-N) exposure, leading to noticeable changes in both behavioral and histopathological bioindicators. The aim herein was to determine the vulnerability of *S. lermae* to $NO_3$-N exposure at realistic concentrations registered in freshwater ecosystems in central Mexico where the species was historically reported. Offspring of *S. lermae* were chronically exposed during 60 days to concentrations of 5, 10 and 20 mg $NO_3$-N/L, with 2 mg $NO_3$-N/L used as the reference value (control). Survival rate, feeding behavior, aquatic surface respiration, body growth, scaled mass index, immature red blood cells, as well as histopathological changes in branchial, hepatic and gonadal tissues were evaluated. Additionally, this study analyzed water quality in freshwater ecosystems where *S. lermae* presently persists. The results showed decreased survival as $NO_3$-N concentration increased, as well as increased feeding latency, aquatic surface respiration and histological damage in the gills and liver. These organs showed differential sex-dependent responses to $NO_3$-N exposure; females were more sensitive than males. In the ovaries, a decreased density of stage III oocytes was associated with increased $NO_3$-N concentrations. No changes were observed in body growth and number of immature red blood cells. Concentrations recorded in the three freshwater ecosystems that *S. lermae* inhabit were below 2 mg $NO_3$-N/L. Together, the results could explain why the species has disappeared from more contaminated

freshwater ecosystems where $NO_3$-N levels exceed 5 mg/L. Moreover, the study warns about the risks of increasing $NO_3$-N concentrations in the current sites where the species lives.

# INTRODUCTION

Nitrogen is one of the more widespread pollutants in aquatic ecosystems (*Galloway et al., 2004*) which appears as ammonia ($NH_3$), ammonium ($NH_4$), nitrite ($NO_2$) and nitrate ($NO_3$. *Camargo & Alonso, 2006*). Because $NO_3$ shows high persistence and solubility, it is the most stable and common form of inorganic nitrogen in natural ecosystems (*Baker et al., 2017*). Elevated $NO_3$ concentrations in freshwater ecosystems are toxic to aquatic organisms (*McGurk et al., 2006*; *Edwards & Guillette, 2007*; *Gomez Isaza, Cramp & Franklin, 2018*; *Gomez Isaza, Cramp & Franklin, 2021*). Although the precise physiological mechanisms underlying these effects have not been widely described, the main consequence of $NO_3$ exposure is a reduction of oxygen transport produced by methemoglobinemia and gill damage (*Yang et al., 2019*). The branchial tissue protects itself from $NO_3$ by cell hyperplasia, hyperemia and lamellar fusion (*Monsees et al., 2017*; *Villa-Villaseñor et al., 2022*). These processes reduce the available surface for gas exchange, promoting behavioral and physiological compensatory mechanisms, such as aquatic surface respiration (ASR) and increased release of splenic red blood cells (*Gomez Isaza, Cramp & Franklin, 2021*). Chronic exposure to elevated $NO_3$ concentrations leads to permanent gill damage, decreased body growth and endocrine disruption (*McGurk et al., 2006*; *Hamlin et al., 2008*; *Edwards, Miller & Guillette, 2005*; *Kellock, Moore & Bringolf, 2018*). Nitrate sensitivity is species-specific and could be related to body size (*Hamlin, 2006*; *Villa-Villaseñor et al., 2022*), feeding habits (*Weis, Smith & Zhou, 1999*), reproductive strategy (*Camargo, Alonso & Salamanca, 2005*), detoxifying abilities (*Gomez Isaza, Cramp & Franklin, 2018*; *Villa-Villaseñor et al., 2022*) and sex (*Cano-Rocabayera et al., 2019*). Moreover, $NO_3$ sensitivity could be associated with the ontogenic stage during which exposure occurs (*Hamlin, 2006*; *Adelman et al., 2009*).

The Goodeinae subfamily comprises a group of endemic fish representative of central Mexico, characterized by inner fertilization, sex dimorphism, complex courtship and matrotrophy (*Wourms, Grove & Lombardi, 1988*; *Iida et al., 2019*). Most Goodeinae species are in critical conservation status (International Union for Conservation of Nature *International Union for Conservation of Nature (IUCN), 2023*). Although several anthropic impacts could explain their diminished distribution and survival, pollution is a critical factor that continues to threaten the maintenance of freshwater ecosystems (*Lyons et al., 2019*; *Lyons et al., 2020*). *Skiffia lermae* is a member of the Goodeinae subfamily whose historical distribution in Zacapu, Yuriria, Pátzcuaro, Zirahuén and Cuitzeo lakes and Lerma river drainages has been limited to a few freshwater ecosystems in Zacapu, Pátzcuaro and Cuitzeo

drainages (*Lyons et al., 2019*). Nowadays, this species is classified as threatened (*Secretariat of Environment and Natural Resources of Mexico, 2010*) and endangered (*International Union for Conservation of Nature (IUCN), 2023*). Previous work showed that *S. lermae* is sensitive to $NO_3$. The median lethal concentration ($LC_{50}$) was the lowest of the four goodeid species studied (*Villa-Villaseñor et al., 2022*) and one of the lowest reported so far for freshwater fish (*Monson, 2022*).

Nitrate-nitrogen ($NO_3$-N) concentrations recorded in springs with low human impact were 0.015 mg/L (*Allan & Castillo, 2007*). Monitoring studies at Zacapu, Pátzcuaro, Yuriria and Cuitzeo lakes have reported 0.08–3.9 mg $NO_3$-N/L, while the Lerma basin showed 0.23–30 mg $NO_3$-N/L (*Espinal-Carreón, Sedeño Díaz & López-López, 2013*; *Pérez-Díaz et al., 2019* and unpublished results). In Mexico, the permissible limit for sewage discharges into freshwater bodies is 15 mg/L of total nitrogen as a daily average (*Secretariat of Environment and Natural Resources of Mexico, 2021*) and there are no regulatory limits for $NO_3$-N. According to the World Health Organization *World Health Organization (WHO)(2003)* guideline, 11 mg $NO_3$-N/L is the permissible limit for human consumption (*Secretariat of Health of Mexico, 2021*). Nevertheless, the negative effects of nitrates on aquatic ecosystems can begin to occur at lower concentrations (*Camargo, Alonso & Salamanca, 2005*).

Given that *S. lermae* has vanished from most freshwater bodies in central Mexico, where pollution levels have risen, and considering a prior study showing its sensitivity to elevated $NO_3$-N concentrations, our hypothesis was that prolonged exposure to sub-lethal, but environmentally plausible $NO_3$-N concentrations, might induce changes in behavioral and physiological indicators even after short exposure periods. To test this hypothesis, offspring of this species were chronically exposed to a reference value (control) and rising nitrate concentrations during 60 days (treatments). Their survival, feeding behavior, ASR, body growth, scaled mass index and number of immature red blood cells, as well as branchial, hepatic and gonadal histology were evaluated. Additionally, because there are no recent reports in freshwater ecosystems where *S. lermae* is currently distributed, this study analyzed $NO_3$-N concentrations on La Mintzita spring, Chapultepec spring and Zacapu lake.

The results of this study could offer valuable insights into the possible causes of the reduction and disappearance of vulnerable and endemic species from freshwater habitats in central Mexico with high $NO_3$-N contamination. Moreover, they may shed light on the physiological responses of this endangered species. These findings provide critical elements for environmental regulation of safer pollutant limits by considering the sensitivity of native species, ensuring their long-term survival.

## MATERIALS & METHODS

### Ethical statements

Sampling and laboratory fish handling protocols were authorized by an Animal Rights Committee under License Number SEMARNAT: SPARN/DGVS/02210/22, following the (*Guide for the Care and Use of Laboratory Animals, 1996*).

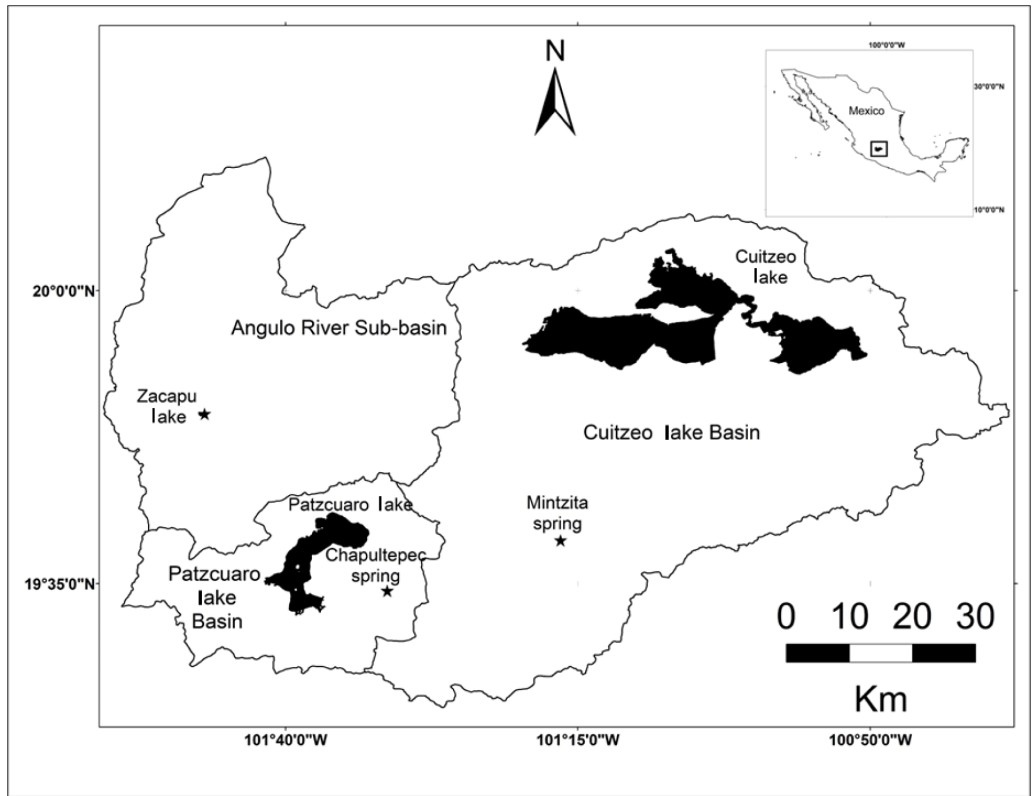

**Figure 1** Geographic location of the three basins and study sites in central Mexico: Zacapu lake (19°49′29″N, 101°46′54″W), La Mintzita spring (19°38′41″N, 101°16′27″W) and Chapultepec spring (19°34′23″N, 101° 31′16″W). Each site is indicated by a star (*Instituto Nacional de Estadística y Geografía (INEGI), 2023*).

## Collection of females and habitat characterization

During 2022, thirty gravid *S. lermae* females were captured from Zacapu lake, central Mexico (19.824193°N; 101.787312°W) with aluminum mesh minnow traps (Gee-minnow-traps® G-40, USA). Females were transported in bags containing water from the collection site and acclimated to laboratory conditions (*Villa-Villaseñor et al., 2022*).

In order to update reference water quality where *S. lermae* remains, Zacapu lake in the Angulo river sub-basin of the Lerma-Chapala basin, La Mintzita in Rio Grande-Cuitzeo basin and Chapultepec spring in Patzcuaro lake basin were evaluated in September 2023, near the end of the rainy season (Fig. 1). The following water parameters were recorded using a multiparameter probe (OAKTON®. Sydney, Australian): hydrogen potential (pH), dissolved oxygen and water temperature.

Water samples were transported to the laboratory to determine the concentrations of ammonium-nitrogen ($NH_4$-N), nitrite-nitrogen ($NO_2$-N) and $NO_3$-N by colorimetric methods at 410 nm, 543 nm and 220 nm, respectively; following the methodologies described in the American Public Health Association (*American Public Health Association (APHA), 2017*).

## Fish maintenance and water quality monitoring in the laboratory

Gravid females were acclimated for 15 days in 120 L aquarium tanks (loading capacity: 6 L per fish) under a 12 h light/darkness photoperiod. Acclimation tanks were loaded using tap water filtered with polypropylene sediment strain, activated carbon, ultrafiltration membrane and ultraviolet light and maintained at $22 \pm 0.5$ °C, pH $7 \pm 0.5$ log units and $7 \pm 0.5$ mg/L dissolved oxygen. Aquariums were artificially enriched to avoid stress and limit fry cannibalism. Females were fed twice a day with commercial flakes (Tetra®) and *Artemia* sp., their health was monitored daily until the birth of their offspring.

At 1–7 postnatal days, a total of 120 *S. lermae* fry were randomly assigned to 12 new 15 L aquariums (10 fish per aquarium; 1.5 L per fish) and acclimated for 3 weeks during which they were fed *ad libitum* with *Artemia* sp. and commercial flakes (Tetra®) according to *Villa-Villaseñor et al. (2022)*. During the entire acclimation and experimental period, a semi-static experimental system was maintained at the same laboratory conditions previously described. Temperature, dissolved oxygen, pH, salinity, total dissolved solids and electrical conductivity were recorded every 72 h with a multi-parameter probe (YSI EXO2. Ohio, USA). In addition, $NH_4$-N, $NO_2$-N and $NO_3$-N were measured as aforementioned (*American Public Health Association (APHA), 2017*).

## Chronic toxicity assays

At three weeks of age, the offspring's body mass and length were measured using an analytical balance (OHAUS™ Adventurer, $d = 0.0001$ g, China) and a Vernier caliper (Thermo Fisher Scientific™ S/N 1366162, EE USA), respectively. The recorded values for each morphological parameter were then averaged among all fish per aquarium to determine the initial body mass and length. Each of the 12 aquariums (three replicates per condition) were randomly assigned to a control ($NO_3$-N concentration close to 2 mg/L, un-amended from the water facility) or treatment condition (5, 10 and 20 mg $NO_3$-N/L). Thus, 30 offspring (10 per aquarium) were exposed in each of the four groups. The $NO_3$-N concentrations used in this study (5, 10 and 20 mg/L) were obtained from concentrations recorded in central Mexico (*Trujillo-Cárdenas et al., 2010*; *Espinal-Carreón, Sedeño Díaz & López-López, 2013*; *Pérez-Díaz et al., 2019*) using sodium nitrate ($NaNO_3$, 97%, Sigma-Aldrich) according to a stoichiometric calculation (*Dutra et al., 2020*). Fifty percent of the aquarium's water was replaced every 72 h, and $NO_3$-N concentrations were adjusted according to nominal values.

The experiment began 1 week after the morphological measures were taken and lasted 60 days. Once $NO_3$-N was added and dissolved (except in the control condition), organism survival was determined every 24 h. The criterion to confirm mortality was an absent response to a mechanical stimulus. The organisms were fed daily with commercial flakes (Tetra®) at 10% of the biomass per aquarium (*Rueda-Jasso, Delos Santos-Bailón & Campos-Mendoza, 2017*). Web camera recordings (Logitech C505 HD WEBCAM) were used to evaluate the ASR and feeding behavior every 24 h (for 20 min at 9 AM). The ASR was evaluated during the first 5 min of recording, counting the number of organisms distributed 5 cm below the water surface (*Gomez Isaza, Cramp & Franklin, 2021*; *Rosales-Pérez et al., 2022*). Feeding behavior was evaluated during the remaining 15

min of recording, by quantifying the latency to detect food, the number of feeding fish and food consumption (percentage of food consumed by organisms: total consumption, 100%; partial, 50% and without consumption, 0%), according to *Rueda-Jasso, Delos Santos-Bailón & Campos-Mendoza (2017)*.

At the end of the experiment, surviving fish were euthanized using an overdose of benzocaine anesthetic (75 mg L-1. *Rueda-Jasso, Delos Santos-Bailón & Campos-Mendoza, 2017*) and their body mass, as well as standard length were recorded as described above (final body mass and final standard length). Absolute growth, specific growth rate and scaled mass index (body condition) were evaluated according to the following equations:

(a)  Absolute growth [g] = final body mass - initial body mass.
(b)  Specific growth rate [% per day] = 100 (([final body mass / initial body mass] ∧[1 / time-days])-1) (*Crane, Ogle & Shoup, 2020*).
(c)  Scaled mass index = final body mass i ((Lo / final standard length i) ∧bSMA), where Lo is the arithmetic mean final standard length for the study population and bSMA is the scaling exponent estimated by the SMA regression of final body mass on final standard length (*Peig & Green, 2009*).

## Hematological and histological analysis

Eighteen fish per control and treatment condition (three males and three females per replicate; except for 20 mg $NO_3$-N/L, where only six males and five females survived) were randomly selected for hematological and histological analysis. A blood sample was taken from the base of the caudal fin to quantify the number of immature red blood cells (*Torres-Bugarín et al., 2007*). A total of 2,000 cells per organism were quantified in a bright field microscope at 1,000x magnification (Leica DM3000).

For histological analysis, fish were fixed in Bouin solution (Sigma) at 4 °C for 48 h and processed following *Villa-Villaseñor et al. (2022)*. Briefly, fish were longitudinally sectioned at 5 μm and stained with hematoxylin and eosin (*Cano-Rocabayera et al., 2019*). For morphometric gill analysis, the thickness of primary lamellae, secondary lamella length and width, as well as the inter-lamellar distance were used to calculate the proportion of total epithelia available for gas exchange (PAGE $_{Tot}$). This per the equation: PAGE $_{Tot}$ [%] = (PAGE * PAGE $_W$)/100, where PAGE [%] = 100 * (mean secondary lamella length/[mean thickness of primary lamellae + mean secondary lamella length]), and PAGE$_W$ [%] = 100 * (mean inter-lamellar distance/[mean inter-lamellar distance + mean secondary lamella width]) (*Maggioni et al., 2012*). Lower PAGE$_{Tot}$ index values mean less available epithelium. To obtain the liver damage tissue index (LDTI), nuclear density (nD = number of nuclei/$mm^2$) and nuclear area (nA = area of hepatocyte nuclei in $mm^2$) were quantified. The average value of controls (nDc and nAc) was used as a reference to calculate a relative value for each parameter. The absolute value of the difference to 1, multiplied by 0.5 was used. Finally, the values were integrated into the following index: LDTI = (nD*1/nD$_c$) + [| 1 −(nA*1/nA$_c$)| x 0.5] (*Villa-Villaseñor et al., 2022*). Values above the control group indicate alterations related to proliferative and hypertrophic processes, while values below the control group denote degenerative damage (*Macirella et al., 2023*). Gonadal analyses were performed only in females. Oocyte morphology was used to determine the ovarian stage
(Table S1) according to *Uribe et al. (2006)*, *Uribe, Grier & Parenti (2012)* and *Tinguely, Lange & Tyler (2019)*. Because oocytes are large cells that occupy most of the ovary, total oocytes with cytoplasmic oil droplets (stage III, previtellogenic oocytes) were quantified in medial ovarian sections. The percentage of atresia was quantified using total stage III oocytes in the same section (*Uribe et al., 2006*, Table S2).

## Data analyses

A total of 120 fish were used in the four experimental groups. Sample size was determined following the recommendations of the Organisation for Economic Co-operation and Development (*Organisation for Economic Co-operation and Development, 2013*): 210 fish for early-life chronic toxicity tests (*Oris, Beñanger & Bailer, 2012*) and according to a previous study (*Villa-Villaseñor et al., 2022*). Aquariums were randomly assigned to control and treatment conditions using the Sample function (*R Core Team, 2023*).

Data normality and equal variance were analyzed using Shapiro–Wilk and Levene tests. Ecosystem freshwater and aquarium water quality parameters were compared using ANOVA or Kruskal–Wallis analyses followed by post-hoc tests, according to the statistical assumptions. The survival rate was calculated by the Kaplan–Meier method followed by the log-rank test. Food consumption was analyzed by a Chi-square test. The rest of the response variables were analyzed by generalized linear mixed models, which included $NO_3$-N concentration as a fixed factor and aquarium id as a random factor (*Harrison et al., 2018*). To evaluate if the effects of $NO_3$-N concentrations were different along the exposure period, behavior models also included time (divided in three periods of 20 days), as well as the interaction of $NO_3$-N concentration with time as fixed factors. Body growth and scaled mass index, as well as hematological and histological models included sex and the interaction of $NO_3$-N concentration with sex as fixed factors. The latter were performed using five males and five females per experimental condition, because at 20 mg $NO_3$-N/L only five females survived. Generalized linear mixed models were performed using Poisson (the number of active fish and ASR), Gaussian (body growth and scaled mass index), Gamma (LDTI and oocytes density) and Beta distributions (PAGE $_{TOT}$ and the percentage of atresia). The best model was chosen by Akaike's information criterion, analysis of deviance ($D^2$) and visual inspection of residuals. Statistical significance was defined as $p \leq 0.05$. All results are shown as mean ± standard deviation. Analyses were done with R (*R Core Team, 2023*) using the following packages: lme4 (*Bates et al., 2015*), DHARMa (*Hartig, 2022*), glmmTMB (*Brooks et al., 2017*), gamlss (*Rigby & Stasinopoulos, 2005*), ggplot2 (*Wickham, 2016*), survival (*Therneau, 2022*) and survminer (*Kassambara, Kosinski & Biecek, 2021*).

# RESULTS

## Freshwater ecosystem and aquarium water quality

The water habitat of *S. lermae*, registered during September 2023, presented a tendency for alkalinity (pH = 7.81 ± 0.93 log units), a mean temperature of 20.6 ± 2.06 °C and mean dissolved oxygen 6.11 ± 0.66 mg/L. Significative differences were found in temperature, pH, dissolved oxygen, $NO_3$-N and $NH_4$-N among each distribution site (Table 1). The

**Table 1** Water quality (mean ± standard deviation) measured in the freshwater ecosystems where *Skiffia lermae* populations persist nowadays (*Lyons et al., 2019*).

|  | La Mintzita | Chapultepec | Zacapu |
|---|---|---|---|
| pH (log units) | $7.79 \pm 0.564^c$ | $6.94 \pm 0.243^b$ | $8.99 \pm 0.166^a$ |
| Temperature (° C) | $20.22 \pm 1.163^b$ | $19.13 \pm 1.319^b$ | $22.93 \pm 1.458^a$ |
| Dissolved oxygen (mg/L) | $6.26 \pm 0.299^a$ | $5.74 \pm 0.122^b$ | $6.45 \pm 1.064^{ab}$ |
| $NO_3$-N (mg/L) | $1.73 \pm 0.028^a$ | $0.81 \pm 0.047^b$ | $0.19 \pm 0.094^c$ |
| $NO_2$-N (mg/L) | $<0.0002^a$ | $<0.0002^a$ | $<0.0002^a$ |
| $NH_4$-N (mg/L) | $0.004 \pm 0.002^a$ | $0.004 \pm 0.0001^b$ | $0.003 \pm 0.0002^{ab}$ |

**Notes.**
Different letters indicate significant differences between treatments.

**Table 2** Water quality (mean ± standard deviation) measured in experimental $NO_3$-N treated aquariums.

| Water quality | 2 mg/L $NO_3$-N (reference condition) | 5 mg/L $NO_3$-N | 10 mg/L $NO_3$-N | 20 mg/L $NO_3$-N |
|---|---|---|---|---|
| $NO_3$-N (mg/L) | $2.084 \pm 0.451^d$ | $5.087 \pm 0.613^c$ | $9.314 \pm 0.531^b$ | $18.762 \pm 1.111^a$ |
| $NO_2$-N (mg/L) | $0.006 \pm 0.003^a$ | $0.011 \pm 0.010^a$ | $0.011 \pm 0.011^a$ | $0.009 \pm 0.005^a$ |
| $NH_4$-N (mg/L) | $0.0028 \pm 0.0002^a$ | $0.0028 \pm 0.0002^a$ | $0.0029 \pm 0.0005^a$ | $0.0028 \pm 0.0002^a$ |
| Temperature (° C) | $21.956 \pm 0.805^a$ | $21.892 \pm 0.810^a$ | $21.951 \pm 0.821^a$ | $21.976 \pm 0.797^a$ |
| Dissolved oxygen (mg/L) | $6.385 \pm 1.008^a$ | $6.418 \pm 0.993^a$ | $6.470 \pm 0.889^a$ | $6.587 \pm 0.868^a$ |
| Conductivity (us/cm) | $277.461 \pm 28.200^c$ | $302.233 \pm 23.644^c$ | $346.553 \pm 24.867^b$ | $445.789 \pm 35.984^a$ |
| Total dissolved compounds (mg/L) | $191.692 \pm 20.467^c$ | $208.782 \pm 16.447^c$ | $239.281 \pm 17.526^b$ | $307.500 \pm 23.631^a$ |
| Salinity (ppt) | $0.140 \pm 0.015^c$ | $0.152 \pm 0.012^c$ | $0.176 \pm 0.013^b$ | $0.228 \pm 0.018^a$ |
| pH (log units) | $8.006 \pm 0.275^a$ | $7.982 \pm 0.242^a$ | $7.964 \pm 0.223^a$ | $7.916 \pm 0.213^a$ |

**Notes.**
Different letters indicate significant differences between treatments.

$NO_3$-N concentrations ($X^2_{(2)} = 25.61$; $p \leq 0.001$) and $NH_4$-N ($X^2_{(2)} = 9.36$; $p = 0.009$) were lower in Zacapu than in La Mintzita or Chapultepec (Table 1).

Aquarium water quality was similar for all experimental conditions except for those parameters linked to $NO_3$-N concentration such as salinity, conductivity and total dissolved compounds (Table 2).

### Survival rate

Survival diminished as $NO_3$-N concentrations and exposure time increased, but only 20 mg $NO_3$-N/L was different from the control condition ($X^2_{(3)} = 48.8$; $p \leq 0.001$). At the end of the experiment 97%, 90%, 80% and 37% of the population survived at 2, 5, 10 and 20 mg $NO_3$-N/L, respectively (Fig. 2).

### Feeding behavior and aquatic surface respiration (ASR)

An effect of $NO_3$-N concentration and exposure period was observed on the latency to food consumption. At 5 and 10 mg $NO_3$-N/L, an increased latency for food detection was observed during the last period, while at 20 mg $NO_3$-N/L an increment was observed from the first observation period. Latency for food detection was reduced as exposure time increased (Fig. 3A and Table S3). No effect was observed on the number of fish actively responsive to food (Fig. 3B and Table S3). Aquatic surface respiration increased as $NO_3$-N

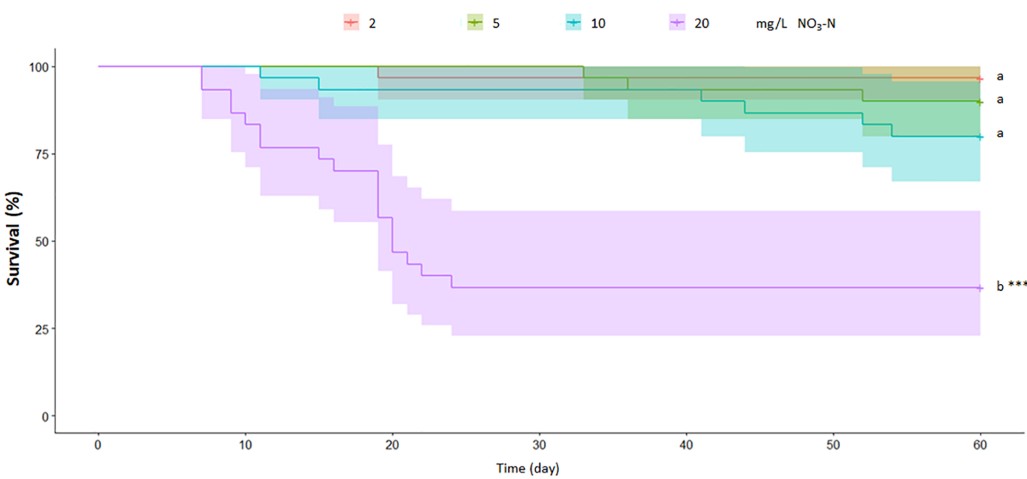

**Figure 2** **Chronic exposure to ecologically relevant NO₃-N concentrations diminished *Skiffia lermae* survival.** Kaplan–Meier survival curves show the survival rate for *Skiffia lermae* fish exposed to 2 (control condition; $n = 30$), 5 ($n = 30$), 10 ($n = 30$) and 20 mg NO₃-N/L ($n = 30$). Log rank tests indicate a significant difference at 20 mg/L. Shading = 95% confidence intervals. ***$p \leq 0.001$.

concentration and exposure time augmented (Fig. 4 and Table S3). Food consumption was decreased only at 20 mg NO₃-N/L compared to the control group ($X^2_{(6)} = 20.395$; $p = 0.002$. Figure 5).

## Body growth and scaled mass index

No differences in body growth (absolute growth, specific growth rates) or the scaled mass index were observed at the different NO₃-N concentrations, sex or their interaction (Table S4).

## Hematological and histological analyses

Nitrate-nitrogen exposure, sex or their interaction did not affect the number of immature red blood cells (Table S5). Nitrate-nitrogen concentration diminished the gill PAGE $_{Tot}$ index up to 5 mg NO₃-N/L. An effect of sex was evident at 10 and 20 mg NO₃-N/L. Females showed a lower PAGE$_{Tot}$ index than males (Fig. 6 and Table S5).

Liver damage tissue index also showed differences from the control group at the distinct NO₃-N concentrations. At 5 and 10 mg NO₃-N/L, LDTI increased, while at 20 mg NO₃-N/L, this indicator diminished. An effect of sex was observed only at 5 mg NO₃-N/L; females showed the highest values (Fig. 7 and Table S5).

Because females were more sensitive than males, ovarian histology was evaluated. Females from all experimental groups were at the same previtellogenic gonadal stage III. Nitrate-nitrogen at 10 and 20 mg/L decreased the density of stage III oocytes compared to the control group. No differences were observed in the percentage of atresia (Fig. 8 and Table S5).

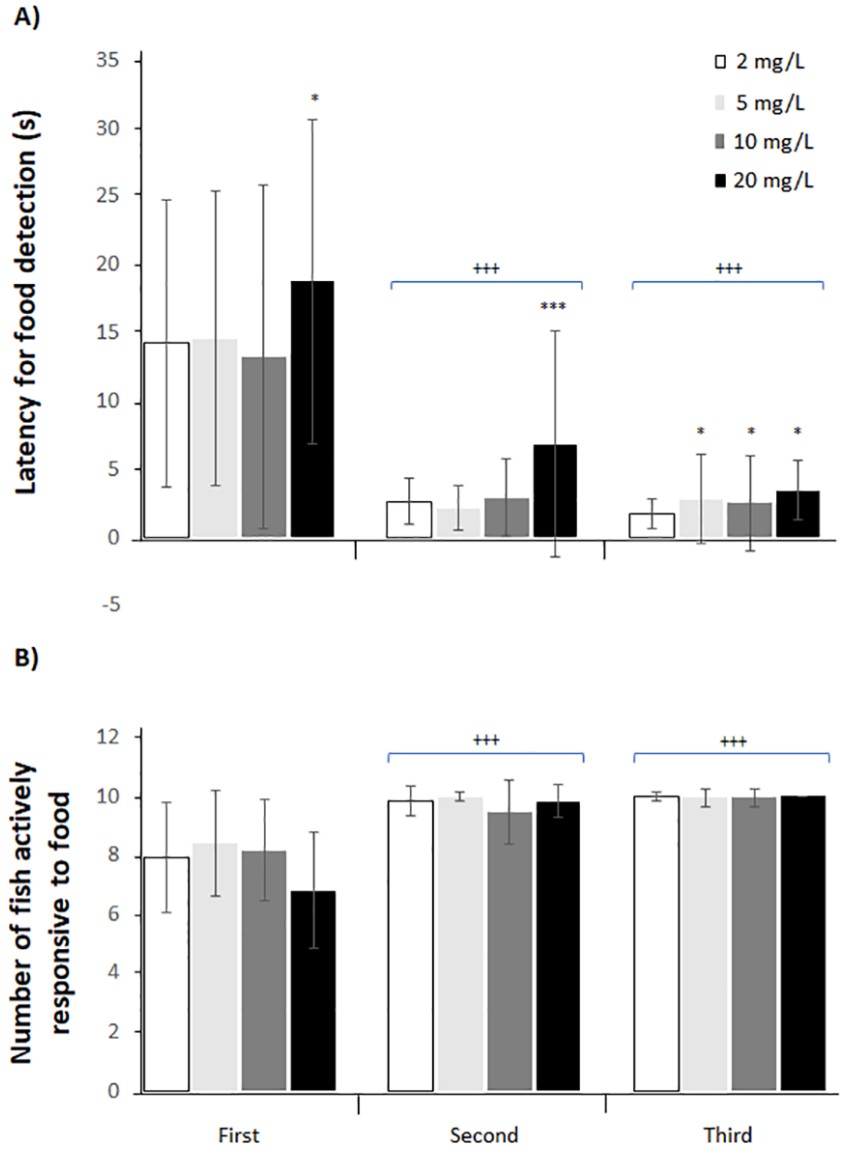

**Figure 3 Ecologically relevant NO₃-N concentrations caused alterations in *Skiffia lermae* behavior.** The time required for food detection (latency/s, A) as well as the number of fish actively responsive to food (B) during the first (1–20 days), second (21–40 days) and third exposure periods (41–60 days) at 2 (control; $n = 30$), 5 ($n = 30$), 10 ($n = 30$) and 20 mg NO₃-N/L ($n = 30$). $^{+++}p \leq 0.001$, first period *vs.* second and third periods; $^{*}p \leq 0.05$, 2 *vs.* 5, 10 and 20 mg NO₃-N/L; $^{***}p \leq 0.001$, 2 *vs.* <20 mg NO₃-N/L. Blue brackets indicate treatments grouped by second and third periods, respectively.

## DISCUSSION

*Skiffia lermae* is severely threatened and is listed as endangered (*International Union for Conservation of Nature (IUCN), 2023*). Its distribution has been reduced more than 60% compared to its historical records. Healthy populations are now found in a few freshwater ecosystems in central Mexico (*Domínguez-Domínguez et al., 2008*; *Lyons et al., 2019*). Previous research has shown that *S. lermae* is highly sensitive to NO₃-N (*Villa-Villaseñor*

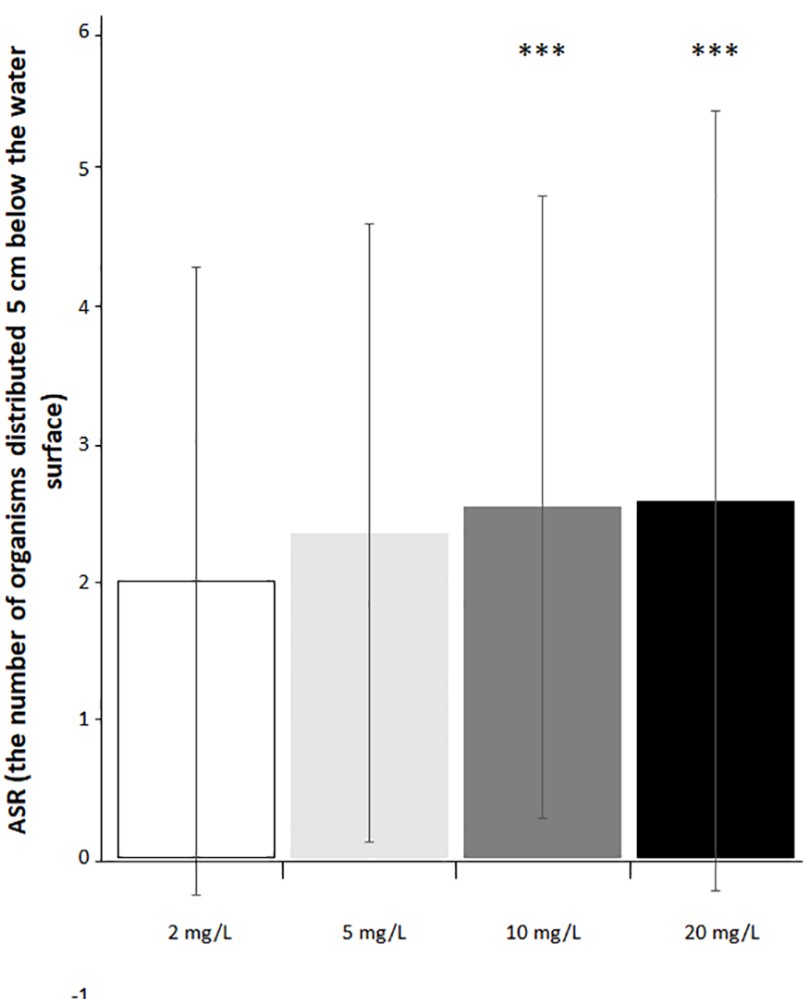

**Figure 4 Ecologically relevant NO$_3$-N concentrations caused alterations in aquatic surface respiration (ASR).** The number of the organisms distributed five cm below the water surface exposed to 2 (control), 5, 10 and 20 mg NO$_3$-N/L. *$p \leq 0.05$, 2 *vs.* 5 mg NO$_3$-N/L; ***$p \leq 0.001$. 2 *vs.* 10 and 20 mg NO$_3$-N/L.

*et al., 2022*). Thus, high and persistent NO$_3$-N concentrations in freshwater ecosystems could threaten their survival. To test this hypothesis, we conducted a study to evaluate the impact of exposure to realistic NO$_3$-N concentrations on several physiological parameters. Our results indicate that exposure to these concentrations can alter behavior, increase gill and liver histopathological indicators, reduce the density of germinal cells in the ovary and decrease fish survival. We also found that females are more sensitive to NO$_3$-N than males. Additionally, we show that concentrations recorded in the three freshwater ecosystems that *S. lermae* inhabit were below 2 mg NO$_3$-N/L. Our study shows the deleterious effects of realistic NO$_3$-N concentrations on a non-model fish wild species and emphasizes the importance of revisiting the NO$_3$-N limits in freshwater ecosystems to ensure the survival of threatened endemic populations.

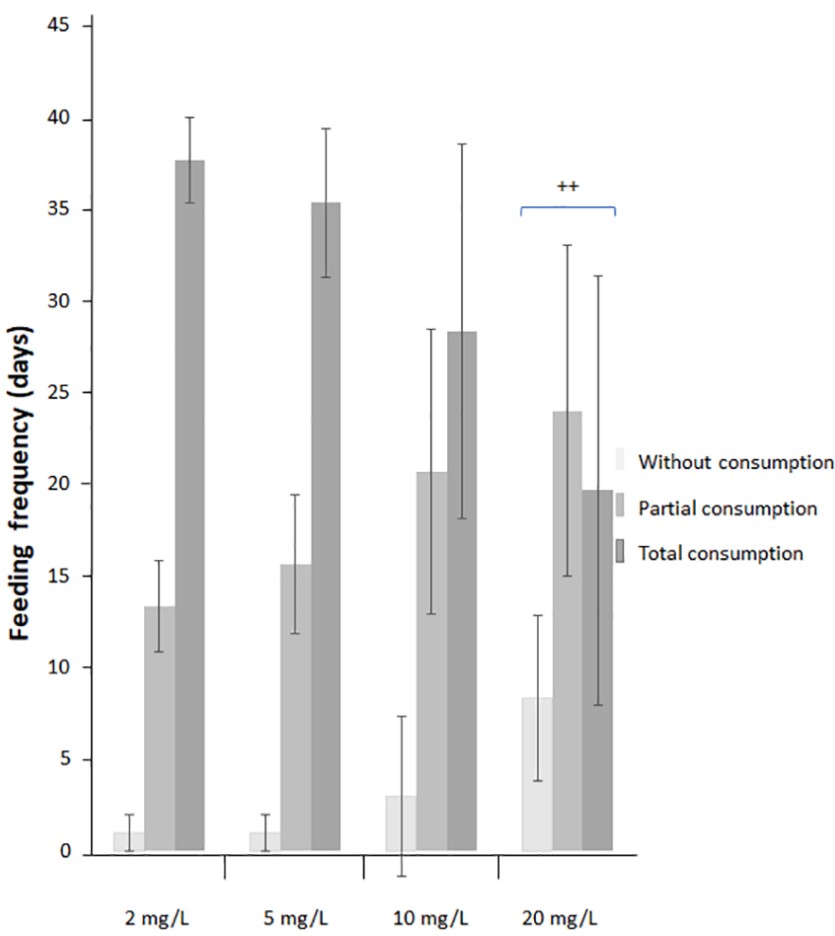

**Figure 5** **Environmentally relevant NO₃-N concentrations decrease food consumption in *Skiffia lermae*.** The feeding frequency of fish exposed to 2 (control condition), 5, 10 and 20 mg NO₃-N/L. **$p \leq$ 0.01, 2 *vs.* 20 mg NO₃-N/L. Blue brackets group food consumption at 20 mg NO₃-N/L.

## Chronic exposure to realistic NO₃-N concentrations diminished *Skiffia lermae* survival

The distribution of *Skiffia lermae* has been reduced during the last 20 years (*Domínguez-Domínguez et al., 2008*; *Lyons et al., 2019*) probably due to increased anthropogenic contamination. Nitrates constitute a pollution source that reduces the survival of sensitive fish species (*Gomez Isaza, Cramp & Franklin, 2020*). Herein, *S. lermae* offspring were exposed to 5, 10 and 20 mg NO₃-N/L, as well as a control concentration (2 mg/L). This value was similar to the highest concentration registered at freshwater ecosystems where *S. lermae* still remains. It is noteworthy that this value was measured during the rainy season, a period characterized by heightened eutrophication in waterways due to runoff carrying nitrogen and phosphorus from agricultural and fossil fuel sources into freshwater ecosystems. Subsequent investigations should track water quality in freshwater environments inhabited by *S. lermae* over an annual cycle.

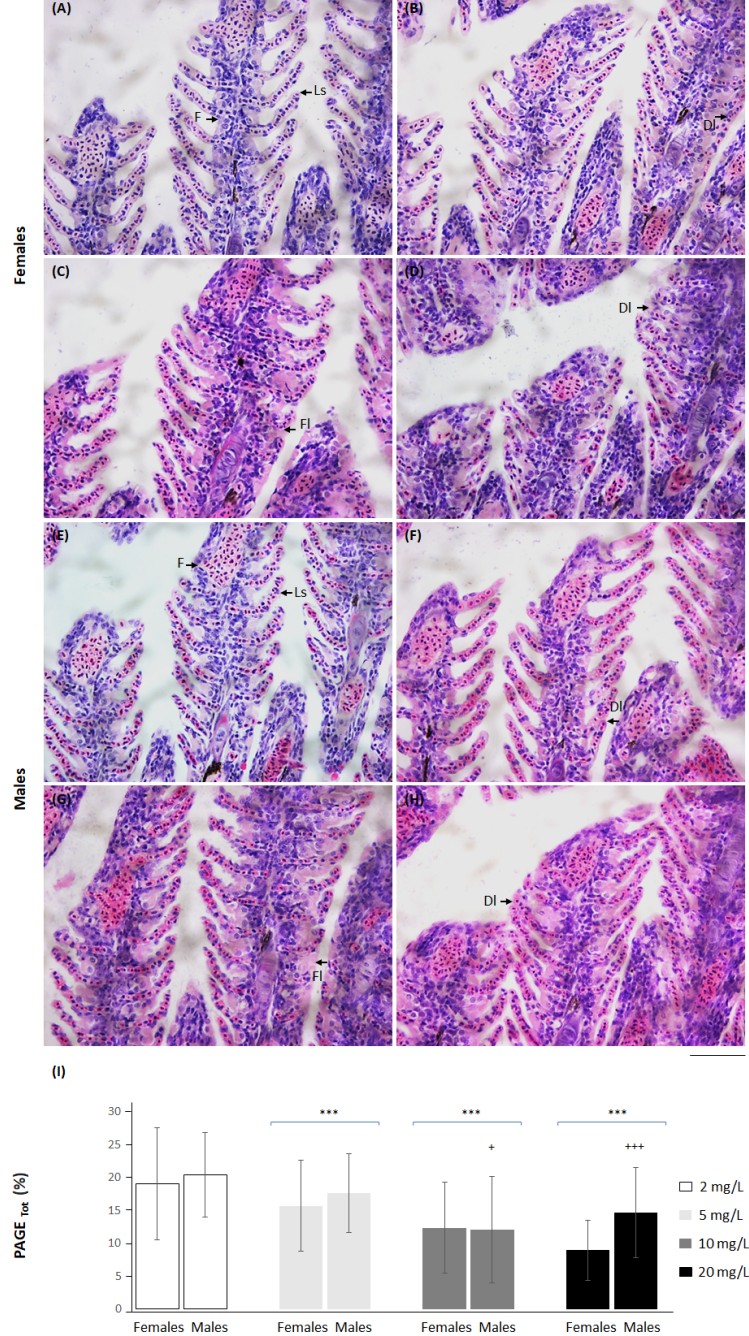

**Figure 6** *Skiffia lermae* **fish show gill histomorphometric alterations at ecologically relevant NO₃-N concentrations.** Representative gill histological sections from females (A–D) and males (E–H) exposed to 2 (control condition; $n = 5$, A and E), 5 ($n = 5$, B and F), 10 ($n = 5$, C and G) and 20 mg NO₃-N/L concentrations ($n = 5$, D and H). DI, decreased interlamellar distance; F, filament; FL, fusion of secondary lamellae; Ls, secondary lamellae. Scale bar = 50 μm. The graph (I) shows the percentage of total epithelia available for gas exchange (PAGETot) in females and males exposed to 2, 5, 10 and 20 mg NO₃-N/L. $^{+}p \leq$ 0.05, females *vs.* males; $^{+++}p \leq 0.001$ females *vs.* males; ***$p \leq 0.001$, 2 *vs.* 5, 10 and 20 mg NO₃-N/L. Blue brackets indicate females and males grouped by concentrations of 5, 10 and 20 mg NO₃-N/L, respectively.

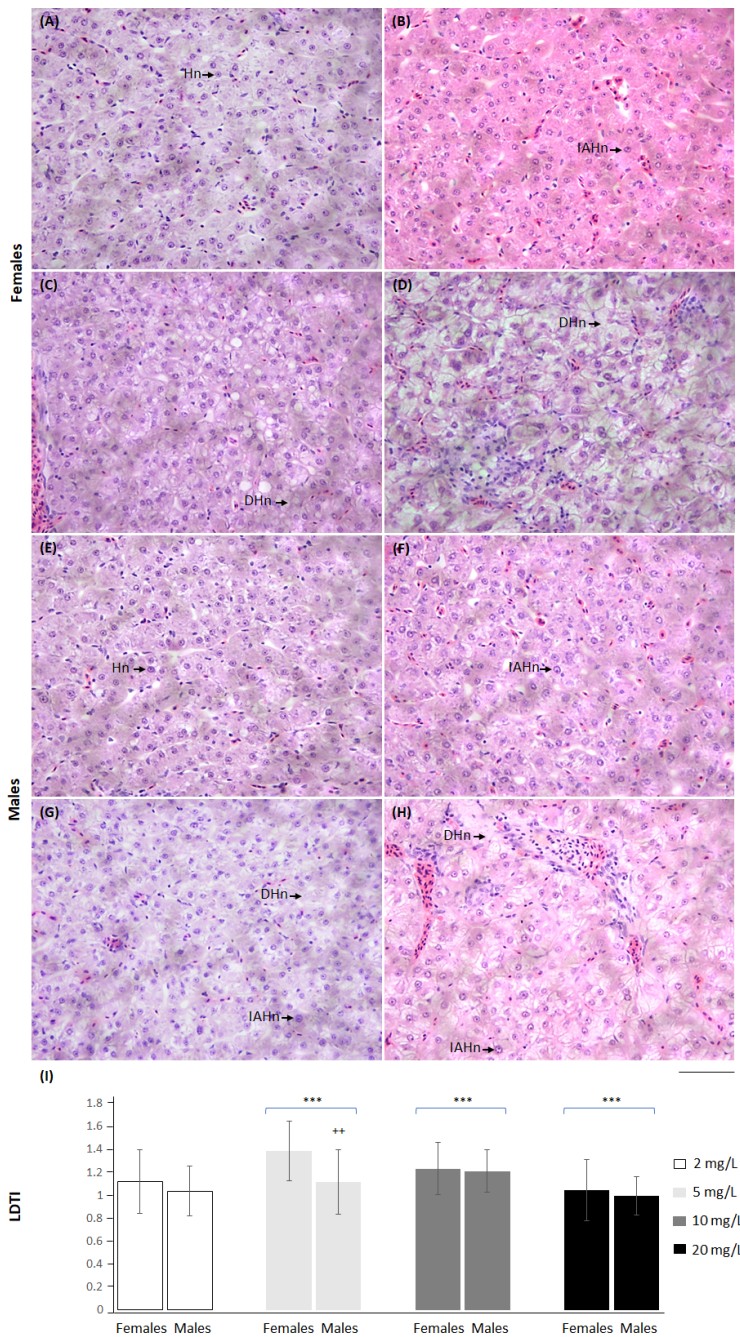

**Figure 7** *Skiffia lermae* **fish show liver histomorphometric alterations at ecologically relevant NO$_3$-N concentrations.** Representative liver histological sections from females (A–D) and males (E–H) exposed to 2 (control condition; $n = 5$, A and E), 5 ($n = 5$, B and F), 10 ($n = 5$, C and G) and 20 mg NO$_3$-N/L concentrations ($n = 5$, D and H). DHn, decrease in hepatocyte nuclei; Hn, hepatocyte nucellus; IAHn, increase in the nuclear area of hepatocytes. Scale bar = 50 µm. The graph (I) shows the liver damage tissue index (LDTI) in females and males exposed to 2, 5, 10 and 20 mg NO$_3$-N/L. $^{++}p \leq 0.01$, females *vs.* males; $^{***}p \leq 0.001$, 2 *vs.* 5, 10 and 20 mg NO$_3$-N/L. Blue brackets indicate females and males grouped by concentrations of 5, 10 and 20 mg NO$_3$-N/L, respectively.

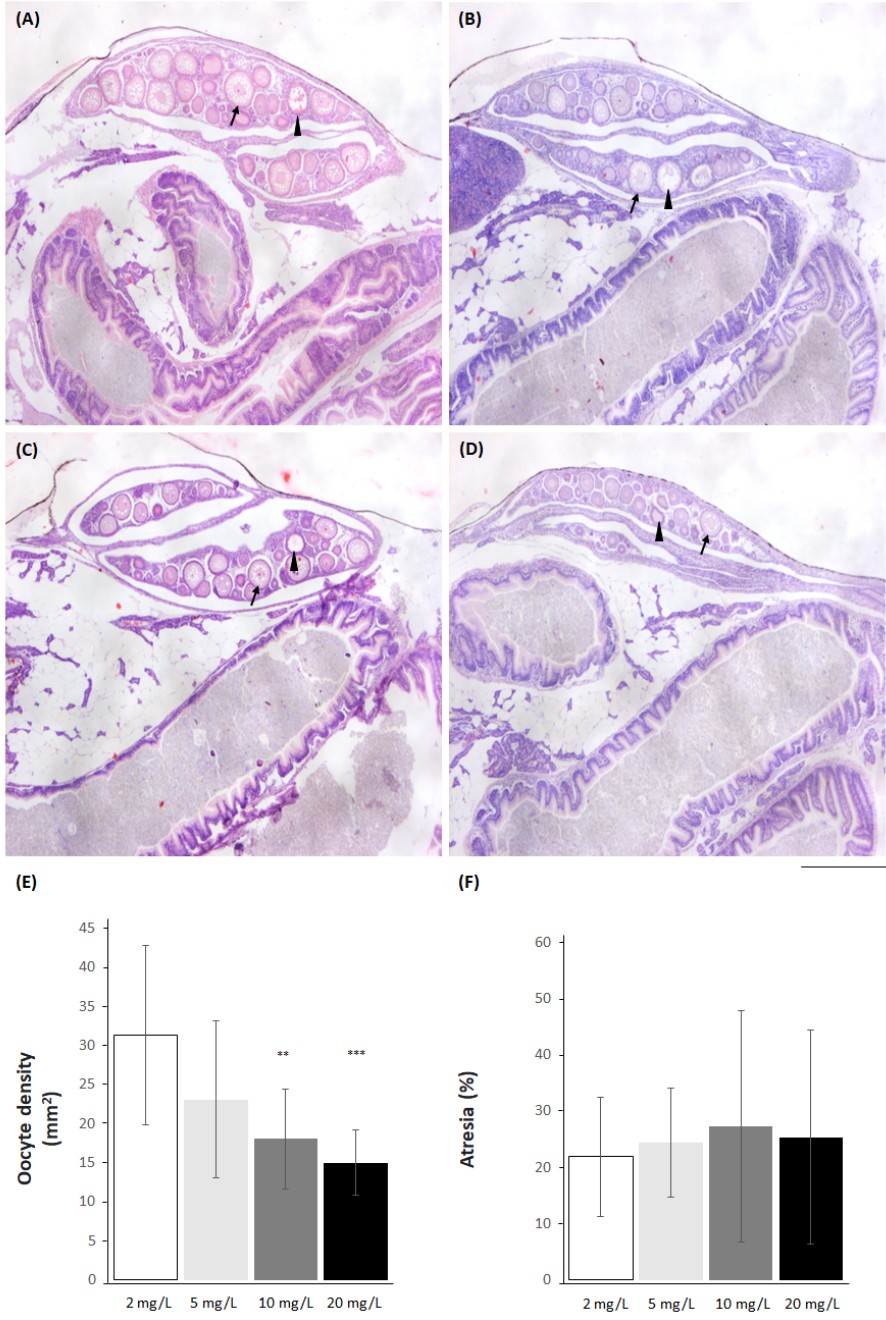

**Figure 8** *Skiffia lermae* **fish show ovarian histomorphometric alterations at ecologically relevant NO₃-N concentrations.** Representative ovarian histological sections from females (A–D) exposed to 2 (control condition; $n = 5$, A), 5 ($n = 5$, B), 10 ($n = 5$, C) and 20 mg NO₃-N/L ($n = 5$, D). Arrows, stage III oocytes; arrow heads, atretic oocytes. Scale bar = 500 μm. Graphs show oocyte density (mm2, E) and atresia (%, F) in females exposed to 2, 5, 10 and 20 mg NO₃-N/L. **$p \leq 0.01$, 2 *vs.* 10 mg NO₃-N/L; ***$p \leq 0.001$ 2 *vs.* 20 mg NO₃-N/L.

At 20 mg NO₃-N/L *S. lermae* survival decreased during the first 20 days of exposure, only 37% of all fish remained at the end of the experiment (60 days). At 10 mg NO₃-N/L, 80% of the fish population survived. Considering that the permissible limit for

sewage discharges into freshwater is 15 mg/L of total nitrogen (*Secretariat of Environment and Natural Resources of Mexico, 2021*) and 11 mg $NO_3$-N/L for human consumption (*Secretariat of Health of Mexico, 2021*), our results suggest that these parameters are well above the critical threshold for sensitive fish species and endanger their long-term survival. This idea is supported by evidence showing that uncontrolled discharges of municipal wastewater played a critical role in water quality degradation of freshwater ecosystems (*De la Lanza-Espino & García Calderón, 1995*; *Gagné et al., 2002*; *López-López, Sedeño Diaz & Perozzi, 2006*). This deterioration caused a population decline and local extinctions of fish fauna in the Cuitzeo basins, which belong to La Mintzita spring (*Soto-Galera et al., 1999*; *Lyons et al., 2019*). Therefore, if the impacts of anthropization continue, there is a severe risk of increasing $NO_3$-N concentrations to intolerable levels for *S. lermae.* Monitoring programs at these sites are necessary to guarantee the survival of endemic species.

### The lowest $NO_3$-N concentration caused alterations in *Skiffia lermae* feeding behavior without modifying their body growth

The results showed that prolonged exposure to $NO_3$-N (at all the concentrations tested) increased the latency to feeding behavior. At 20 mg NO3-N/L feeding was altered starting from the 1st observation period (see methods). Exposure to 20 mg $NO_3$-N/L also diminished the amount of ingested food. No change was observed in the number of actively responsive fish. The observed differences concerning the second and third periods in latency and actively responsive fish could be explained by acclimation to new social interactions and diminished survival during the first period. Previous studies showing that fish respond to new environments, conspecifics and population density support this observation (*Crane et al., 2018*).

Fish feeding behavior is determined mainly by visual and chemical signals and depends on a functional motor system (*Hara, 2006*; *Morais, 2017*). The fact that *S. lermae* spends more time close to the water surface suggests sensory or motor system alterations. However, no differences were observed in the amount of food consumed between the control group and fish exposed to 5 and/or 10 mg $NO_3$-N/L. No differences were observed in body size between the control and treatment conditions. Nevertheless, it is important to note that alterations in feeding behavior at treatment conditions were evident until the final exposure period. Thus, it is likely that prolonged exposure to $NO_3$-N may result in varying growth rates, as was shown in *Salvelinus namaycush* (*McGurk et al., 2006*). Alternatively, the $NO_3$-N concentrations used in our study may not have been sufficiently high to induce changes in fish body size, as previously reported in other studies (*Kellock, Moore & Bringolf, 2018*). In the case of *S. lermae* individuals exposed to 20 mg $NO_3$-N/L, diminished food consumption could be related to fatigue and low oxygen consumption due to the decreased epithelium available for gas exchange (see below), as previously suggested (*Cano-Rocabayera et al., 2019*).

### *Skiffia lermae* increased their aquatic surface respiration and showed gill histomorphometric alterations at the lowest $NO_3$-N concentrations

Aquatic surface respiration is an early indicator of low oxygen consumption caused by diminished hemoglobin or branchial damage (*Gomez Isaza, Cramp & Franklin, 2021*).

Hemoglobin is a blood protein that transports oxygen from the gills to the rest of the body. Toxic $NO_3$-N concentrations oxidize hemoglobin to methemoglobin, a protein unable to transport oxygen efficiently (*Camargo & Alonso, 2006*; *Yang et al., 2019*; *Presa et al., 2022*). The initial mechanisms to increase oxygen exchange include splenic erythrocyte release and their accumulation in gills (hyperemia), which increases the proportion of immature red blood cells circulating in the blood (*Gomez Isaza, Cramp & Franklin, 2021*). Persistent toxic concentrations produce sequential gill compensatory and degenerative mechanisms, such as hyperplasia and lamellar fusion, respectively (*Antunes et al., 2017*). Herein, no changes were observed in immature red blood cells between control and treatment groups at the end of the experiment. However, the gill histomorphometric index showed a diminished $PAGE_{Tot}$ (the proportion of total epithelial available for gas exchange) at all treatment conditions. These results suggest that even the lowest $NO_3$-N concentration (5 mg) was high enough to trigger structural compensatory and degenerative branchial alterations. As previously shown, blood-associated compensatory mechanisms could occur during early $NO_3$-N exposure (*Gomez Isaza, Cramp & Franklin, 2020*; *Presa et al., 2022*). At 20 mg $NO_3$-N/L, females showed a greater reduction in $PAGE_{Tot}$ than males, which suggests that females are more susceptible to nitrates than males.

### *Skiffia lermae* showed progressive liver histomorphometric alterations as $NO_3$-N concentrations increased

The LDTI (which includes hepatocyte nuclear area and density) was increased at 5 and 10 mg, but decreased at 20 mg $NO_3$-N/L. An increased LDTI reflects cell proliferation and nuclear hepatocyte hypertrophy, early compensatory mechanisms that raise metabolic and transcriptional activity oriented to increase detoxification processes (*Bangru & Kalsotra, 2020*). A decreased LDTI denotes degenerative changes associated with reduced hepatocyte activity and death (*Villa-Villaseñor et al., 2022*). The results suggest that $NO_3$-N/L concentrations up to 5 mg induce liver changes, which could be reversible if environmental conditions are better, while 20 mg $NO_3$-N/L could produce irreversible changes. Again, *S. lermae* females showed an increased hepatocyte nuclear area and density at 5 mg $NO_3$-N/L. Given that *S. lermae* is a viviparous species, reproduction may require increased liver activity to synthesize molecules, such as vitellogenin, as previously indicated (*Iida et al., 2019*). Thus, a trade-off to preserve reproductive function may allow liver damage. Previous studies suggest that sex-associated differences in liver alterations could be related to the differential absorption, distribution, metabolism and excretion of toxic substances (*Flores, Manautou & Renfro, 2017*). To evaluate if augmented liver alterations in females may be related to reproductive functions, germinal cells were analyzed in ovarian sections.

### Realistic $NO_3$-N concentrations diminished reproductive indicators in *Skiffia lermae* females

*Skiffia lermae* is a viviparous species in which fertilization and embryonic development occur in the ovary (*Wourms, Grove & Lombardi, 1988*; *Iida et al., 2019*) and sexual maturity is reached approximately at 29.5 $\pm$ 5.7 mm body length (*Ramírez-García et al., 2021*). Females in each reproductive period have between six to 23 offspring, which is the lowest

fertility value, compared with other goodeines species in Zacapu lake (*Ramírez-García et al., 2021*). Here, *S. lermae* females showed an average $21.73 \pm 0.28$ mm body length, which suggests that they are sexually immature and supports the absence of vitellogenic oocytes.

Atresia is a physiological process where degeneration and reabsorption of oocytes are observed, which makes it possible to recover part of the energy and components invested during follicular maturation (*Uribe et al., 2006*). Atresia mainly affects follicles that contain oocytes with yolk (*Corriero et al., 2021*). However, its appearance has been reported in atretic follicles without yolk in fish, affecting previtellogenic oocytes at a stage where cortical lipid alveoli are observed (*Corriero et al., 2021*). A progressive decrease in stage III oocytes was observed at 10 and 20 mg $NO_3$-N/L with no changes in the percentage of atresia. Because stage III oocytes will mature to become fertilized and develop an embryo, a decreased number of oocytes will reduce the fertility index. This could contribute to diminished *S. lermae* populations in the long term. These results support previous studies showing that nitrates can be considered endocrine disruptors for females in lecithotrophic (*Pimephales promelas. Kellock, Moore & Bringolf, 2018*) and incipient matrotrophic species (*Gambusia holbrooki. Edwards, Miller & Guillette, 2005*). *Edwards, Miller & Guillette (2005)* demonstrated a correlation between nitrate concentrations and decreased indicators of reproductive investment, including embryo dry weight and reproductive activity in wild mature females. Similarly, *Kellock, Moore & Bringolf (2018)* found elevated vitellogenesis in both males and females, along with heightened 11-ketotestosterone levels in males associated with increased nitrate concentrations. A recent study demonstrated varying sensitivities between male and female *G. holbrooki* exposed to different $NO_3$-N concentrations, with males exhibiting greater sensitivity than females (*Cano-Rocabayera et al., 2019*). The findings of this last study contradict those of the present research, which indicates that females exhibit greater sensitivity than males. Given that *G. holbrooki* and *S. lermae* follow distinct developmental patterns linked to reproductive investment, it is plausible that matrotrophy imposes a heightened energetic demand on females of *S. lermae*, as previously suggested (*Trexler & De Angelis, 2003*).

## CONCLUSIONS

This study shows that exposure to ecologically relevant freshwater $NO_3$-N concentrations promoted compensatory and degenerative changes in *S. lermae* even after brief exposure periods. Chronic exposure over 60 days resulted in decreased *S. lermae* survival associated with behavioral, branchial and hepatic alterations. Additionally, exposure to different $NO_3$-N concentrations diminished early indicators of reproductive investment in *S. lermae* females. These findings underscore the importance of reevaluating guidelines governing wastewater discharge into freshwater ecosystems to safeguard the health of vulnerable endemic fish species.

### Funding

This work was supported by the Coordinación de la Investigación Científica-UMSNH to Esperanza Meléndez-Herrera and Omar Domínguez-Domínguez; Chester Zoo Garden; Association Beauval Nature Pour la Conservation et la Recherche; The Mohammed Bin Zayed Species Conservation Found; Goodeid Working Group; American Livebearers Association to Omar Domínguez-Domínguez; and Ivette Marai Villa-Villaseñor was a Ph.D. fellow from CONACYT (grant no. 743476). The funders had no role in study design, data collection and analysis, decision to publish, or preparation of the manuscript.

### Grant Disclosures

The following grant information was disclosed by the authors:
Coordinación de la Investigación Científica-UMSNH.
Chester Zoo Garden.
Association Beauval Nature Pour la Conservation et la Recherche.
The Mohammed Bin Zayed Species Conservation Found.
Goodeid Working Group.
American Livebearers Association.
CONACYT: 743476.

### Competing Interests

The authors declare there are no competing interests.

### Author Contributions

- Ivette Marai Villa-Villaseñor conceived and designed the experiments, performed the experiments, analyzed the data, prepared figures and/or tables, authored or reviewed drafts of the article, and approved the final draft.
- Ma. Antonia Herrera-Vargas performed the experiments, analyzed the data, prepared figures and/or tables, and approved the final draft.
- Beatriz Yáñez-Rivera conceived and designed the experiments, analyzed the data, authored or reviewed drafts of the article, and approved the final draft.
- Mari Carmen Uribe analyzed the data, authored or reviewed drafts of the article, and approved the final draft.
- Rebeca Aneli Rueda-Jasso conceived and designed the experiments, authored or reviewed drafts of the article, and approved the final draft.
- Bryan V. Phillips-Farfán analyzed the data, authored or reviewed drafts of the article, and approved the final draft.
- Valentin Mar-Silva performed the experiments, analyzed the data, prepared figures and/or tables, and approved the final draft.
- Esperanza Meléndez-Herrera conceived and designed the experiments, authored or reviewed drafts of the article, and approved the final draft.
- Omar Domínguez-Domínguez conceived and designed the experiments, authored or reviewed drafts of the article, and approved the final draft.

## Animal Ethics

The following information was supplied relating to ethical approvals (i.e., approving body and any reference numbers):

The Mexican authority (SEMARNAT) approved the sampling procedure, as well as maintenance and experimental protocols. (SEMARNAT: SPARN/DGVS/02210/22)

## Field Study Permissions

The following information was supplied relating to field study approvals (i.e., approving body and any reference numbers):

The Mexican authority (SEMARNAT) approved the field techniques and experimental conditions related to the fish during experimentation
(SEMARNAT: SPARN/DGVS/02210/22).

## Data Availability

The raw data are available in the Supplemental Files.

## Supplemental Information

Supplemental information for this article can be found online at http://dx.doi.org/10.7717/peerj.17876#supplemental-information.

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
