# Peer review of "Realistic nitrate concentrations diminish reproductive indicators in Skiffia lermae, an endemic species in critical endangered status"

_PeerJ, doi:10.7717/peerj.17876_

## Round 0.1 · original submission · Major Revisions

Your manuscript needs to be thoroughly revised and many aspects need to be improved. One of the most important is to improve the discussion, for which a more thorough review of the subject should be made to better understand the importance of the results obtained.

In addition, all the reviewers' comments should be taken into account to modify the manuscript accordingly.

**Language Note:** The review process has identified that the English language must be improved. PeerJ can provide language editing services - please contact us at [email protected] for pricing (be sure to provide your manuscript number and title). Alternatively, you should make your own arrangements to improve the language quality and provide details in your response letter. – PeerJ Staff

·

Basic reporting

General reporting of the manuscript is good, except for the quality of the images of histopathology, which can have a better resolution and larger size. Supplementary materials add a significant amount of data to the main manuscript
Minor comments:
lines 58-59, nitrogen appears also as nitrites or nitrates in natural waters. Rephrase please
line 76, I think 'ontogenic' is more used instead of ontogenetic, check please.
lines 95-98: this sentence mixes sewage discharges (a flux) with the concentration of 11 mg/l NO3-N. Rephrase to make it more clear to the reader.

Experimental design

Probably the calculation of growth rates and condition factors need to be revised. Instead of absolute growth rate, it might be more informative to calculate specific growth rate which takes into account the initial mass. Check ‘Use and misuse of a common growth metric: guidance for appropriately calculating and reporting specific growth rate’ by Derek P. Crane, Derek H. Ogle, and Daniel E. Shoup for a correct usage (page 2). Also I assume that the initial weight (W0) was a mean of all fish? Because I assume it was difficult to identify each fish for tracking W0 and Wf. Please clarify this in the text.
Also for calculating Fulton condition factor, the usage of the assumed scaling exponent of 3 is somehow reviewed. Check ‘New perspectives for estimating body condition from mass/length data: the scaled mass index as an alternative method’ by Jordi Peig, Andy J. Green for a condition factor that uses a specific scaling exponent for your species. It might give same results, but it will probably be more accurate.
Minor comments:
line 131: remove 'dissolved O2 saturation (percentage)'
lines 132-134: detail how you measure the nitrogen species, is it a kit, colorimetry?
line 137: add the day/night hour cycle
line 146: add once the photoperiod time, do not repeat if it is the same
lines 147-151: again, do not repeat the parameters measured if they are the same ones as those measured in the stocking tank, to avoid being redundant.
line 157: change unamended for 'Control' or 'reference' and other times it appear in the manuscript from now on. Actually, 2 mg/l NO3-N is low but not a real proper control. I guess it is the concentration of nitrates coming to the water facility as control water. It could be noted, though, that Camargo et al (2005) suggest a maximum value of 2.0 mg/l NO3-N for the most sensistive species, as a reference value for this study.
line 158: change experimental for ‘treatment’ and other times it appear on the text, because experimental conditions are all.
lines 178-180: all specifications already written above.
lines 210-211: information repeated
lines 216-218: avoid informing ‘were evaluated’, but go directly saying the methods to evaluate each.
line 225: these two references are not statistical works, just apply the method. Rephrase or cite a study in statistics suited for this statement.

Validity of the findings

This study adds data on the toxicity of nitrate to other previously published papers on these issue. The main novelty of this paper I think it is the contribution of knowledge on chronic toxicity, which is fairly scarce, and the use of a non-model species from a rather rare used family in toxicity studies.
The section of results is well described and clear. It would be good to clarify whether the measurements of nitrogenous compounds in the field were during the period of most fertilisation in the area. Because a single measurement at the end of the rainy season can give biased conclusions. It would be worth, if possible, to include measurements of other seasons to have a better perspective.
The Discussion section is to me the part of the study that needs some improvement, since a few aspects are very little put in the context of the present knowledge on the toxicity of nitrates in fish. For example, is it possible that the no differences in growth rate can be attributed to a low period between t0 and tf for differences to be seen?
Most importantly, a better discussion on the higher effects of nitrate on females is expected, since it contradicts the previous study of Cano-Rocabayera et al (2019) on the mosquitofish Gambusia holbrooki. Could it be related to the different form of development? Being the mosquitofish lecithotrophic, compared to the matrotrophic Skiffia lermae, could this be a major difference, regarding the time window dedicated to energy allocation for reproduction, and the time exposed to nitrate? Also check ‘Chronic nitrate exposure alters reproductive physiology in fathead minnows’ by Kellock et al (2018) in fathead minnows for sex differences.
The conclusions however are highly hypothetical and reaching further than the controlled experiment described here. Better adjust to the results reported here (e.g. the suggestions regarding the longer exposure periods and the exposure and the effect on the fitness on other species).
See just some minor comments below:
minor comments (Results section)
lines 245 - 246 (and possibly other sections that I missed): add the units for each parameter.
line 247: add the actual p-value when citing ‘significant differences’ or refer to the table with the results. Proceed likewise in other parts of the results section that has similar phrasing.

Minor comments on the discussion section:
lines 298 - 299: quite bold this statement, I can think of a few studies getting animals from the wild and exposing to nitrate in the lab, as you did in the present manuscript.
lines 306 - 308: repeated information
line 317: cite this statement or personal observation i f it is your own, but it lacks explanation if it is your personal statement.
line 349: ‘...oxidizes hemoglobin to methemoglobin...’
line 352 - 353: reference this statement on the increase of red blood cells.

Figure 6 could improve the quality of the pictures and the size to better see the alterations.
Figure 7, same comment regarding the quality of the figures, it seems pixelated when zooming in.

Additional comments

This study is a continuity to the previous work 'Differential sensitivity of offspring from four species of goodeine freshwater fish to acute exposure to nitrates', contributing additional data on the toxicity of nitrates to a non-model Goodeinae species, but this time in a chronic study. Both non-model species and chronic are to me the major contributions of this work. See comments above to address my suggestions, especially on the Discussion section.

Reviewer 2 ·

Basic reporting

This manuscript (manuscript ID #95813) provides a thorough and complete evaluation of the sublethal and lethal effects of nitrate on an endangered fish species. The hypothesised that chronic exposure to sublethal but realistic NO3-N concentrations could cause early alterations in physiological indicators. The study has elegantly tested this hypothesis through a controlled, laboratory experiment on wild-caught (1st generation in laboratory) fish. The data presented is generally clear and informative. The manuscript is also well written and logical. The manuscript is well cited and provides sufficient background into the problem being addressed.

The hypothesis presented is vague. What does “cause early alterations in physiological indicators” mean? What is early – rapid, acute changes? Few rapid measures were made except for behavioural measures.

The figures and tables are generally clear. Though some small changes could be made:
Tables: Data should be presented as mean (+/-) standard deviation and not standard error.
Figures 3 - 8: What do the error bars represent (SE or SD)?
Figures 3 - 8: units should be in the x-axis label and not on the axis itself.
Figure 2 – change x-axis from Time (Day) to Time (day)
Figure 4 – Add units to y-axis
Figure 5 – This figure is unclear. Perhaps a stacked bar chart would better represent the data.

Experimental design

Overall, the study does not provide novel insight into the impacts of nitrate on fish. The results presented have been documented elsewhere. The novelty of this study is in the quantification of these effects on an endangered fish from Mexico. That being said, I believe that manuscript is worthwhile and does present a complete evaluation (behavioural, physiological-toxicological, histology) of the effects of nitrate on this threatened species.

Line 137: Change “Acclimatization” to “Acclimation” here and throughout. Acclimatization refers to physiological changes to field conditions, whereas acclimation refers to physiological changes in response to controlled, laboratory conditions.

Line 157: Change “…an unamended” to “control/reference”.

169 – 176: When was ASR and feeding behaviour quantified? After the 60-day exposure period? Please make this point explicit. The results show that this was measured three times. I think this needs to be added to the methods.

The number of feeding fish and food consumption measures: Food consumption was grouped coarsely into either total consumption, 100%; partial, 50% and without consumption, 0%. Why was this done so coarsely? What does these data actually provide? Similarly, number of fish feeding could be impacted by density-dependent mechanisms among treatments. Given that mortality was not equal among treatments, number of fish feeding cannot be disentangled from treatment effects.

The proportion of total epithelia available for gas exchange (PAGETot) and liver damage tissue index (LDTI) are not described in much detail and, instead, divert the reader to a citation. I would recommend that some description of what they are and how they were calculated be included in the manuscript.

Validity of the findings

For the feeding assay, the control group often consumed all the food provided (~40 out of 60 days). This would suggest to me that they were underfed. Maybe this is why there was no difference in growth and condition among treatments?

The authors conclude by stating that “our results suggest reassessing the guidelines that establish wastewater discharges into freshwater.” However, no details on current guidelines are presented and no indication on what a “safe” nitrate level might be for this species.

---

## Round 0.2 · accepted · Accept

Thank you very much for making the suggested modifications to the manuscript, which can now be accepted for publication in PeerJ.

Thank you for choosing this journal.

·

Basic reporting

The revised manuscript has clearly improved based on my own the suggestions and the ones from the other anonymous reviewer.

The quality of the photographs has greatly improved, although I would suggest of adding in the histological Figures 6 and 7 the scale indication of 50 micrometres under the scale bar, in addition to the caption, because it is a little bit unclear now.

The English is now correct based on my non-native knowledge.

Experimental design

The experimental design was good in my opinion before the resubmission and it continues to be now.

Validity of the findings

The results section now is good with the small improvements suggested, and the discussion, which in my opinion was the weakest section in the previous manuscript, is now better referenced and written in regard to previous findings.